# Bispecific Antibodies for Lymphoid Malignancy Treatment

**DOI:** 10.3390/cancers17010094

**Published:** 2024-12-31

**Authors:** Matteo Bisio, Luca Legato, Filippo Fasano, Corrado Benevolo Savelli, Carola Boccomini, Maura Nicolosi, Elisa Santambrogio, Roberto Freilone, Mattia Novo, Barbara Botto

**Affiliations:** Hematology Division, A.O.U. Città della Salute e della Scienza di Torino, C.so Bramante 88, 10126 Turin, Italyrofreilone@cittadellasalute.to.it (R.F.); mnovo@cittadellasalute.to.it (M.N.)

**Keywords:** bispecific T-cell engagers (BiTEs), bispecific antibodies (BsAbs), B-cell lymphoma (B-NHL), CD20, CD19, CD30, antibody

## Abstract

Bispecific antibodies (BsAbs) are changing the roadmap in the treatment for B-cell lymphomas, especially in the relapsed/refractory setting. BsAbs are a subtype of bispecific T-cell engagers (BiTEs) which are compounds designed to facilitate T-cell-mediated cell death by redirecting T-cells to attack tumor cells. This review examines the mechanism of action; the available published and ongoing data from clinical trials; and the management of therapy-related adverse events for anti-CD20, CD19, and CD30 BsAbs in the treatment of lymphoid malignancies.

## 1. Introduction

At least a third of patients with B-cell non-Hodgkin lymphoma (B-NHL) still require second- or further-line treatment despite recent advances in frontline therapy or the standard salvage approaches [1,2,3,4,5]. CD19-directed chimeric antigen receptor T-cell (CAR-T) therapies are actually changing the outcome of R/R patients with diffuse large B-cell (DLBCL), mantle cell (MCL), and follicular (FL) lymphoma. The long-term data show disease-free outcomes in approximately 30–40% of patients after anti-CD19 CAR-T, including high-risk patients, making it the preferred option for this difficult-to-treat population. Access to CAR-T treatment may be affected by logistic problems, such as the need for hospitalization, geographic factors, and toxicity-related adverse events (AEs), in particular, cytokine release syndrome (CRS) and immune effector cell-associated neurotoxicity syndrome (ICANS) [6,7,8,9,10]. Bispecific antibodies (BsAbs) are molecular immunotherapies, administrable in the outpatient setting, designed to bind to and activate effector T-cells and drive them into tumor-cell antigens. Compared to CAR-T, they show a similar cellular-dependent cytotoxicity, as well as comparable, but generally less severe, immune-related side effects. 

The development of BsAbs throughought the years has involved experimentation with different molecular structures in order to overcome production and performance challenges previously not seen in ordinary monoclonal-antibody-based compounds [11].

The role of BsAbs in lymphoid malignancy treatment has rapidly evolved in recent years, with trials investigating their use as single agents or in combination in both relapsed and frontline curative-intent settings. In this review, we provide a general overview of recently completed or ongoing trials using BsAbs in patients with R/R B-NHL and classical Hodgkin’s lymphoma (cHL), including single-agent results, emerging combinations, safety data, and novel constructs

## 2. Anti-CD20 Antibodies

### 2.1. Glofitamab

Glofitamab is a CD20 × CD3-targeting bispecific IgG1-based antibody conceived for B-NHL treatment and is currently being studied in several therapeutic settings. In 2023, it was approved for the treatment of R/R DLBCL by the US Food and Drug Administration (FDA) and the European Medicines Agency (EMA) with a fixed-duration treatment of 12 cycles [12,13].

Glofitamab is a full-length humanized BsAb for intravenous (IV) administration with a flexible linker, two domains binding CD20 that utilize the obinutuzumab epitope and one domain binding CD3 (at a 2:1 ratio) [Figure 1]. It has strong affinity for CD20, a prolonged half-life, and a silenced crystallizable fragment (Fc), which consists of an altered Fc region unable to mediate effector functions [14]. Its elevated binding affinity theoretically grants it good efficacy in combination strategies or in sequential therapies with other monoclonal antibodies (mAbs). 

The first infusion of glofitamab is preceded by a single dose of obinutuzumab, and a dose escalation is required before the first cycle in order to reduce CRS [Table 1] [15,16]. Steroid premedication with dexamethasone, routinely used with other bsAbs, appears to further reduce the risk of CRS [17].

Glofitamab exhibits a manageable toxicity profile both as a monotherapy and in combination with other drugs or after CAR-T cell therapy. The most common include CRS, hematological toxicity, and infections. Initial studies reported CRS rates ranging from 20% to 80%, with limited cases of severe events (grade (G) ≥ 3 of 5% or less) [15,18,19,20,21].

Other observed adverse effects included neurological events, typically ICANS (8–15% overall, 3–3.2% G ≥ 3), tumor flare, and tumor lysis syndrome [15,18]. Interestingly, glofitamab appears to cause fewer infections compared to other similar compounds as shown by Fares et al. [22] with 38% of infections globally (1% grade ≥ 3). 

The glofitamab single agent revealed consistent efficacy in heavily pretreated DLBCL, achieving complete response (CR) rates of 36.8–39% and overall response rates (ORR) of 52–53.8%. The median duration of response (mDOR) of 18.4 months and the median duration of CR (mDOCR) was 26.9 months. Notably, glofitamab maintained its efficacy in DLBCL patients who failed CAR-T cell therapy, with CR rates of 36.4% and median progression-free survival (mPFS) and median overall survival (mOS) of 4.9 and 17.6 months, respectively, based on a median follow-up (mFU) of 9.6 months [15,18,23]. In some of these patients with detectable peripheral CAR-T cells, glofitamab infusion led to an expansion of these cells, while, in other cases, no effect was observed [19]. 

Real-world data showed lower response rates in less selected and more pretreated R/R DLBCL patients, with CR rates of 16–23% and ORR of 21–56% ORR [24,25,26]. However, better results were reported in a real-life setting in China by Song et al. [27], where a cohort of B-NHL patients achieved an ORR of 66.7% and a CR rate of 51.9%.

**Table 1 cancers-17-00094-t001:** Glofitamab, mosunetuzumab, and epcoritamab approved dosing schedules.

Glofitamab [12,13]	Cycle (Q21)	Day	Dose		Infusion Duration
	Cycle 1	D1	Obinutuzumab 1000 mg		
		D8	1st step-up dose	2.5 mg	4 h
		D15	2nd step-up dose	10 mg	4 h
	Cycle 2–12	D1	Full dose	30 mg	2 h
Mosunetuzumab [28,29]	Cycle (Q21)	Day	Dose		Infusion Duration
	Cycle 1	D1	1st step-up dose	1 mg	4 h
		D8	2nd step-up dose	2 mg	
		D15	60 mg		
	Cycle 2	D1	60 mg		2 h
	Cycles 3+	D1	30 mg		
Epcoritamab [30,31,32]	Cycle (Q28)	Day	Dose		
	Cycle 1	D1	1st step-up dose		0.16 mg
		D8	2nd step-up dose		0.8 mg
		D15	First full dose		48 mg
		D22	48 mg		
	Cycle 2–3	D1, D8, D15, D22	48 mg		
	Cycles 4–9	D1, D15	48 mg		
	Cycles 10+	D1	48 mg		

Glofitamab has also been tested in other R/R B-NHL settings, revealing even greater efficacy. In a phase I trial involving patients with R/R FL, glofitamab achieved an ORR of 61.9% and a CR rate of 52.4%, with a mPFS of 11.8 months [15]. The trial by Morschhauser et al. [21], in which patients received glofitamab with or without obinutuzumab as a premedication, showed higher efficacy with ORR and CR rates up to 100% and 73.7%, respectively. In this setting, CRS was reported in 66% to 79% of patients, while 40% experienced neurological events.

Glofitamab has also been tested in patients with R/R MCL in a phase I–II trial. Most patients were refractory to their first line of therapy and/or their most recent prior therapy. Promising results were presented at EHA 2024, with a global ORR of 85% and CR rate of 78.3%. Interestingly, among patients previously treated with bruton tyrosine kinase inhibitors (BTKis), the ORR and CR rates were 74.2% and 71%, respectively, with an mPFS of 8.6 months, an mDOR of 12.6 months, and an mOS of 21.2 months. These results are encouraging when compared to the global population of the study, which achieved an mPFS of 16.8 months, an mDOR of 16.2 months, and an mOS of 29.9 months [33,34]. The study included two cohorts with different obinutuzumab pretreatment doses (1000 mg or 2000 mg). The cohort exposed to a higher dose of obinutuzumab experienced reduced CRS rates and fewer cases of G 4 CRS. Globally, ICANS rates were 5–13.5% [34]. 

Currently, the GLOBRYTE study is an ongoing phase III, open-label, multi-center, randomized, controlled trial evaluating the efficacy and safety of glofitamab monotherapy in patients with R/R MCL. This study compares glofitamab against the investigator’s choice of rituximab + bendamustine (BR) or rituximab + lenalidomide (R-Len) [35].

#### Glofitamab in Combined Modalities

Glofitamab is currently being investigated in combination with other compounds in both first-line and subsequent lines of therapy [Table 2]. 

In combination with polatuzumab vedotin (an anti-CD79b monoclonal antibody conjugated with the microtubule inhibitor monomethyl auristatin E), glofitamab achieved an ORR 78% and a CR rate of 56% as a third or later line of treatment in a cohort of over a hundred patients with R/R DLBCL and high-grade B-cell lymphoma (HGBCL), including cases post CAR-T therapy. The 12-month duration of CR was 59%. With an mFU of over 12 months, the mPFS was 10.4 months, while the mOS was not reached [36]. The 12-month analysis of the phase Ib study investigating glofitamab in combination with rituximab plus cyclophosphamide, doxorubicin, vincristine, and prednisone (R-CHOP) in previously untreated DLBCL reported impressive results, with an ORR of 92.9% and a CR rate of 83.9%. These promising outcomes were accompanied by reassuring safety data: 41% neurological events but no ICANS (mostly peripheral neuropathy and paresthesia) and 6 cases of CRS (all G 1–2), a third of which needed tocilizumab. Other common toxicities included infections (mostly G 1–2, 3 cases G 5) and neutropenia (24 cases G 3–4) [37]. These encouraging results were further validated by Falchi et al. [38] in a phase II study involving a high-risk population of untreated DLBCL patients defined by elevated levels of circulating tumor DNA (ctDNA): the end-of-treatment ORR and CR rates were 93.3% and 80%, respectively. The recent STARGLO trial compared glofitamab combined with gemcitabine and oxaliplatin (Glo-GemOx) to the same chemotherapy regimen paired with rituximab in R/R DLBCL transplant-ineligible patients. The experimental combination yielded a 33.2% increase in the CR rate and improved survival compared to the standard arm with an mOS of 25.5 months (vs. 12.9 months) and an mPFS of 13.8 months (vs. 3.6 months). The experimental arm showed higher rates of treatment discontinuation due to AEs but limited CRS rates [39].

A chemo-free combination of glofitamab with rituximab and polatuzumab vedotin is currently being tested in untreated DLBCL patients who are not eligible for immunochemotherapy. While efficacy results are not yet available, preliminary safety data suggest a favorable profile, with no ICANS or severe CRS reported [40].

Glofitamab has also been paired with new compounds such as RG6333, a CD19xCD28 BsAb, which demonstrated enhanced T-cell effector function in ex vivo studies and animal models. These results were further improved with a triple combination that included another agent targeting CD19x4-1BBL, which resulted in better immune activation and antitumor efficacy overall than the glofitamab single agent [41]. Another promising combination involves mapliparcept, an antibody-like fusion composed of a CD47-binding domain, derived from SIRPalfa, linked to an Fc domain. This design enhances phagocytosis and antitumor activity via the inhibition of CD47-mediated signaling. The ongoing phase Ib–II study investigating glofitamab with mapliparcept appears promising [42].

### 2.2. Mosunetuzumab

Mosunetuzumab is a humanized IgG1-based CD20 × CD3 BsAb [Figure 1] with an altered Fc that does not bind the complement or Fc gamma receptor. It has a single rituximab-like binding site for CD20 and a single binding site for CD3. Similar to other CD20 × CD3 BsAbs, the first cycle of mosunetuzumab requires a dose-escalation schedule conceived to reduce the incidence of CRS [Table 1].

Mosunetuzumab was initially tested in patients with R/R FL, revealing a consistent efficacy. The phase I–II study included patients with different subtypes of B-NHL, categorized into indolent non-Hodgkin lymphoma (iNHL), predominantly FL, and aggressive non-Hodgkin lymphoma (aNHL), predominantly DLBCL. The latest update of the study reported an ORR of 65.7 and 36.4% and a CR rate of 49.3 and 21.7% in iNHL and aNHL, respectively. The mPFS was 12.2 months for iNHL and 1.4 months for aNHL, while the mOS was not reached and 9.4 months, respectively. Generally, better responses were observed in third-line treatment compared to later lines and in older patients or those who did not experience disease progression within 24 months after frontline therapy (POD24). However, no survival differences were observed between POD24 and other patients [43]. Based on early results from this study, mosunetuzumab received FDA and EMA approval in 2022 for the treatment of R/R FL after at least two prior lines of systemic therapy [28,29]. The long-term follow-up of R/R FL patients confirmed these efficacy results, showing an mPFS of 24 months, an mDOR of 35.9 months, and an mDOCR that was not reached [44].

In terms of safety, the pivotal phase I–II trials reported a CRS incidence of 27.4–44% (G 3–4 1–2%), limited neurological events and the most common G 3–4 events represented by neutropenia (25.4–27%), hypophosphatemia (15.2–17%), hyperglycemia (2–8%), and anemia (8–9.1%) [45,46]. In this trial, mosunetuzumab was also tested as a subcutaneous (SC) formulation, which showed a favorable toxicity profile, with low CRS rates (27%, all G 1–2) and rare ICANS (3%, all G 1), alongside an efficacy comparable to the IV formulation [47]. 

Despite the results obtained in R/R FL, mosunetuzumab has shown limited efficacy in aNHL such as R/R DLBCL and transformed FL (trFL), with ORR and CR rates of 42% and 23.9%, respectively, and an mPFS of just over 3 months [48]. However, the CRS incidence was lower in this setting (26.1%). Schuster et al. [49] reported similar response rates in a heterogeneous group of patients with R/R FL, DLBCL, trFL, and post CAR-T patients.

As with other CD20 × CD3 BsAbs, mosunetuzumab treatment poses a significant infection risk. A recent retrospective analysis on a single-center cohort treated with mosunetuzumab revealed that, alongside bacterial infections, many viral infections occurred during treatment (54% of the infectious event observed) [50]. Another report highlighted that most infections were not related to neutropenia and were more frequent during the first four cycles. In particular, 9% of patients required immunoglobulin (Ig) support and Ig recovery in patients achieving CR at the end of treatment took nearly 2 years to obtain a 50% increase in B-cells and IgM. However, the IgG and IgA recovery remained below 40% after 30 months [51]. Interestingly, no significant correlation was observed between CD20 expression and mosunetuzumab activity. Most patients in the pivotal trial maintained their CD20 expression during treatment, but a minority experienced a reduced expression after exposure to the drug. Mechanisms such as reduced transcription, the acquisition of truncating mutations, or variants altering the therapeutic binding epitope may influence the activity of anti-CD20 agents, including mosunetuzumab [52].

Mosunetuzumab is also being studied as a frontline treatment for elderly DLBCL patients unfit for intensive chemotherapy. Preliminary results from an ongoing phase I–II trial reported a best ORR of 56% and a CR rate of 43%, with an mDOCR of 15.8 months and manageable toxicity [53].

Enticing results have also been observed in the frontline treatment of FL and marginal zone lymphoma (MZL) in an ongoing single-center study focusing on efficacy and safety. At mid-enrollment, the data show an ORR of 100% and a CR rate of 83%, with no CRS exceeding G 1, no ICANS, and limited hematological and non-hematological AEs [54]. Similar results in terms of efficacy and tolerability were reported in the phase II MORNINGSUN study conducted on first-line, low-tumor-burden FL [55]. Mosunetuzumab also demonstrated efficacy in high-tumor-burden FL, as shown by the preliminary results of the phase II trial by Falchi et al. [56]: the ORR was 96% and the CR rate was 81%, with a comparable efficacy in patients with bulky disease and high-standard-uptake-value (SUV) lesions. 

Fixed-duration, single-agent mosunetuzumab has also been tested in a limited cohort of patients with R/R Richter’s syndrome, achieving an ORR of 40% and a CR rate of 20%. Half of the patients in CR maintained their response for more than 20 months at data cut-off, while the other half proceeded to a hematopoietic stem-cell transplant. Toxicities were manageable, and the observed activity warrants further investigations of mosunetuzumab in this challenging population [57].

#### Mosunetuzumab in Combined Modalities

Mosunetuzumab has demonstrated its potential as a strong partner in combination therapies [Table 2]. The combination of mosunetuzumab and the immunomodulatory agent lenalidomide (mosu-len) represents an intriguing chemo-free option due to the potential synergistic effect of the two drugs. This regimen exhibited a favourable toxicity profile with encouraging anti-lymphoma activity in R/R FL. CRS was limited to G 1–2, and no AEs leading to discontinuation were observed. The CR rate in this study was 77%, although the population was limited, with only ten patients enrolled [58]. A phase III randomized trial comparing mosu-len to R2 for R/R FL patients is ongoing (NCT04712097). Mosu-len is also under investigation in a phase Ib–II trial in untreated FL, administered for 12 cycles using SC mosunetuzumab. Preliminary data from the first 37 patients enrolled showed an ORR of 88.9% with 81.5% CR. A biomarker analysis suggested a positive immunomodulating effect on T cells, favoring central/effector memory CD8+ phenotypes [59]. The combination mosu-len is also being investigated in the MARSUN phase III trial, where it is compared with the investigator’s choice of treatment in R/R MZL [60].

Mosunetuzumab has also been combined with CHOP in a phase II trial for untreated DLBCL patients. The best ORR was 87.5% with a CR rate of 85% and a 2-year PFS of 65%. The mDOR was not reached, while the 2-year DOCR was 70.9%, considering an mFU of 32 months. The combination was relatively well-tolerated with the most common G 3–4 AEs being neutropenia (65%), anemia (30%), and febrile neutropenia (20%). CRS occurred in 60% of cases, mostly G 1–2 during cycle 1, and 12.5% of patients suffered neurological events including cases of ICANS G 2 [61].

Mosunetuzumab combined with polatuzumab vedotin (M-pola) has demonstrated promising activity on large B-cell lymphomas (LBCL). In R/R LBCL, this combination achieved an ORR of 62% and a CR rate of 50%. The mDOCR was not reached, while the mPFS and mOS were 11.4 and 23.3 months, respectively. The safety profile was manageable, with only three patients experiencing G 3 CRS, and five patients experiencing ICANS [62]. The same regimen was tested as frontline therapy in elderly/unfit LBCL patients, obtaining an ORR and a CR rate of 55% and 45%, respectively. Notably, in this phase I–II trial, 41% of patients experienced severe AEs, including 15 fatal events (half of them due to COVID-19 pneumonia) [63]. A second cohort of the same trial is investigating M-pola as salvage therapy for DLBCL patients who achieved a less than partial response (PR) after frontline immunochemotherapy (NCT03677154). Furthermore, M-pola has demonstrated promising efficacy in a cohort of R/R MCL patients after BTKi failure. This combination obtained an ORR and a CR of 75% and 70%, respectively. Remarkably, a significant portion of patients had high-risk features, such as tp53 mutations or aggressive variants, and about one-third had received prior CAR-T therapy. Response rates were consistent across these subgroups, although the median time in the study was generally limited [64].

### 2.3. Epcoritamab

Epcoritamab is a CD20 × CD3 IgG1 full-length BsAb for SC administration [Figure 1] [65,66,67]. Utilizing an ofatumumab-like domain, it binds a specific site of CD20, differing from other approved CD20-targeting BsAbs by engaging both the large and small extracellular loops of the CD20 protein [68,69]. Epcoritamab was approved in 2023 by the FDA and EMA for the treatment of R/R DLBCL, and, in 2024, for R/R FL [30,31,32]. In the dose-escalation studies, no dose-limiting toxicities were observed. Although epcoritamab has been tested at higher dosages, the approved full dosage is 48 mg administered following two dose-escalation steps [Table 1] [65,70]. 

In the dose-escalation phase of the pivotal phase I–II EPCORE NHL-1 study on R/R B-NHLs, epcoritamab demonstrated a particularly favourable toxicity profile, with 59% of patients experiencing G 1–2 CRS and 47% reporting low-grade injection site reactions. The therapeutic activity was promising, with ORR and CR rates of 88% and 38%, respectively, in R/R DLBCL, and 90% and 50%, respectively, in R/R FL [65]. The efficacy of epcoritamab in heavily pretreated R/R DLBCL was confirmed in the dose-expansion cohort of the study, which included 157 patients previously exposed to two or more lines of therapy [66]. The median number of prior lines was 3 (range 2–11), with 61% being primary refractory and nearly 40% having failed prior CAR-T therapy. An ORR of 63.1% and a CR rate of 38.9% were observed, with a mDOR of 12 months. Slightly lower rates of responses were observed in post-CAR-T patients (ORR 54.1% and CR rate 34.4%). Half of the patients experienced CRS of any grade, with 2.5% G ≥ 3. ICANS was rare, occurring in 6.4% of patients (only one case G ≥ 3).

Recently, Linton et al. [71] presented the primary analysis of the phase II dose-expansion cohort of EPCORE NHL-1 in R/R FL. Among 128 R/R FL patients, the ORR was 82% with a CR rate of 62.5%. CRS was reported in approximately 70% of patients, including 2% with G 3. ICANS was limited to 6% of patients with G 1–2 events. Of interest, the analyses included the safety results of the ‘cycle 1 optimization cohort’, where an intermediate dose of epcoritamab 3 mg was administered on day 15. This approach successfully reduced acute toxicities, with CRS observed in 49% of cases (9% G 2, and no G 3 events) and no ICANS reported. 

Epcoritamab has also been tested in patients with Richter’s syndrome transformation, both as a first-line and later-line therapy, demonstrating manageable toxicities and encouraging initial efficacy results in the EPCORE CLL-1 trial [72].

In terms of duration and quality of response, Philips et al. [73] showed that achieving minimal residual disease (MRD) negativity was associated with prolonged responses and improved progression-free survival. MRD negativity was achieved in 33–50% of patients across all subgroups (prior CAR-T, R/R DLBCL, double hit, and triple hit).

#### Epcoritamab in Combined Modalities

Epcoritamab is also being evaluated in combination with immunochemotherapy or other molecules, showing manageable safety profiles and interesting preliminary results [Table 2]. In Arm 4 of the phase Ib–II multi-arm EPCORE NHL-2 trial, the combination of epcoritamab eith rituximab, dexamethasone, cytarabine, and oxaliplatin or carboplatin (R-DHAX/C) was tested as a salvage treatment for transplant-eligible R/R DLBCL patients [74]. This combination obtained an ORR of 86% with a CR rate of 65% overall. We found that 11 out of 27 patients did not proceed to autologous stem-cell transplant (ASCT) after epcoritamab + R-DHAX/C induction but instead continued with the epcoritamab single agent, with the mDOR in this subgroup not reached at the time of data cut-off. All CRS events were low-grade (31% G 1, 10% G 2), and a single G 2 ICANS occurred. 

The combination of epcoritamab with lenalidomide and rituximab (epco-R2) for a fixed duration of 12 cycles was in arm 2 of the EPCORE NHL-2 trial for R/R FL obtaining an ORR of 95% with 73% of CR [75]. The same scheme obtained similar responses (ORR 90% and 69% CR) when adopted as a frontline treatment for FL in arm 6 of the multi-arm study [76]. Based on these results, a randomized phase III trial (EPCORE FL-1) comparing epco-R2 to R2 in R/R FL patients is ongoing (NCT05409066).

Epcoritamab combined with gemcitabine and oxaliplatin was investigated in arm 5 of the EPCORE NHL-2 (arm 5) for R/R DLBCL patients not eligible for transplant [77]. This combination achieved high response rates, with an ORR of 78% and a CR rate of 55%, including consistent results in the subgroup of primary refractory patients. The mDOCR was 13.1 months. However, 13 fatal treatment-emergent AEs were observed in this study.

In arm 8 of EPCORE NHL-2 trial, epcoritamab was combined with dose-reduced R-CHOP (R-mini-CHOP) as a frontline therapy for unfit DLBCL patients. The median age was 81. CRS was observed in 43% of patients, while neutropenia occurred in 32%. We found that 3 out of 28 patients discontinued treatment, and one G 5 AE was reported. Preliminary efficacy results revealed an ORR and a CR rate of 100% and 85%, respectively, among the 20 efficacy evaluable patients [78].

In the phase Ib–II EPCORE NHL-5 multi-arm trial (NCT05283720), epcoritamab is being combined with different agents including lenalidomide, lenalidomide plus ibrutinib, and polatuzumab vedotin with cyclophosphamide, doxorubicin, and prednisone (pola-R-CHP). Preliminary results from arm 1, which evaluated epcoritamab plus lenalidomide in R/R DLBCL, showed promising antitumor activity, with 58% of patients achieving a CR [79].

### 2.4. Odronextamab

Odronextamab is hinge-stabilized, IgG4-humanized CD20×CD3 BsAb [Figure 1] that utilizes an ofatumumab-like domain to bind CD20 [80]. The phase I study enrolled 145 R/R B-NHL patients and had an mFU of 4.2 months. Odronextamab was administered using a dose-escalation approach with initial split doses, weekly infusion until week 12, and a full dose (320 mg for DLBCL and 160 mg for FL and MZL) administered every 2 weeks from week 14 until disease progression or unacceptable toxicity. Across the entire cohort, the ORR and CR rates were 51% and 37%, respectively. A subgroup analysis showed an ORR/CR rate of 91%/72% in FL, 48%/30% in DLBCL after CAR-T, and 53% (all CR) in DLBCL not previously exposed to CAR-T [81,82]. The phase II ELM-2 study found further odronextamab efficacy in R/R DLBCL, with a longer mFU of 26.2 months: 52% ORR (66/127), 31% CR (39/127), and an mDOCR of 17.9 months [83]. Interim results from the R/R FL cohort of the same study confirmed great efficacy: ORR of 80% (102/128) with a CR rate of 72% (92/128), an mPFS of 26.6 months, and an mDOCR of 21.7 months [84]. For both DLBCL and FL groups, the 2-year mDOCR was 48%. The safety profile of odronextamab was consistent with other BsAb, with the most common AEs being infections, hematological toxicities, ICANS, and CRS. However, ICANS and CRS appeared to be more severe compared to other BsAbs. CRS occurred in 61% of patients, with G ≥ 3 in 7%; ICANS occurred in 12% of patients with 3% of G ≥ 3. Eight percent of patients discontinued odronextamab due to AEs [81].

Similar to other CD20 × CD3 BsAbs, odronextamab is being tested in phase III trials as a monotherapy or in combination with immunochemotherapy regimens. These trials compare odronextamab to standard-of-care (SOC) treatments for both iNHL and aNHL, but all studies are recruiting and there are currently no preliminary results available at this time. Combinations with other immunotherapies such as cepilimab, an anti-PD1 drug, and REGN5837, a novel CD22 × CD28 BsAb, has shown interesting co-stimulatory effects when used with odronextamab [85] [Table 2].

### 2.5. Other CD20 × CD3 Antibodies

Plamotamab is a humanized CD20 × CD3 BsAb [Figure 1]. In the phase I study, the incidence of CRS was 72.2%, with all cases being G 1–2. The only G 3–4 AEs observed were related to hematological toxicity. In R/R DLBCL patients, plamotamab achieved an ORR of 47.4% and a CR rate of 26.3%. Better results were seen in R/R DLBCL after CAR-T therapy with an ORR and CR rate of 46.2% and 30.8%, respectively. In R/R FL, plamotamab achieved an ORR and CR rate of 100% and 50%, respectively [86,87].

Imvotamab is the first bispecific IgM CD20 × CD3 [Figure 1], featuring ten high-affinity/avidity binding domains for CD20 and a single binding domain for CD3 [88,89]. In the phase I study, 8 of the 23 evaluable patients obtained a response, with 5 of them attaining a CR. Imvotamab revealed a manageable safety profile compared to other BsAbs available [89]. A pharmacodynamic study by Hernandez et al. [90] highlighted an interesting response of the immune system to Imvotamab, suggesting that its physiological stimulation enhances T-cell effector function without inducing excessive inflammatory cytokine release associated with CRS CRS. 

GB261 is a novel compound designed to retain Fc effector function while reducing CRS incidence [91]. In the first-in-human dose-escalation study conducted in R/R FL and R/R DLBCL patients, GB261 showed promising efficacy with an ORR and CR rate of 73% and 45.5%, respectively, despite a limited FU of 4.5 months. The CRS incidence was 12.8%, all G 1–2, and no ICANS events were reported [92]. 

## 3. Anti-CD19 Antibodies

### 3.1. Blinatumomab

Blinatumomab (MT103, AMG103) is a CD19 × CD3 targeting the bispecific T-cell-engager (BiTE) [Figure 1], currently approved for the treatment of CD19-positive R/R or MRD-positive B-cell-precursor acute lymphoblastic leukaemia (B-ALL) [93]. It consists of two murine-derived single-chain variable fragments (ScFvs) recombinantly joined by a short, non-immunogenic linker of five amino acids [94].

Initial clinical experience in B-NHLs was obtained from three phase I dose-escalation studies conducted in the early 2000s (MT103-1/01-2001, MT103-1/01-2002, and MT103-1/01-2003). Blinatumomab was administered once, twice, or three times weekly via a two- to four-hour IV infusion at doses ranging from 0.75 to 13 μg/m^2^. All three short-term infusion trials were terminated early due to the limited clinical benefit with notable adverse events (AEs) such as neurologic toxicities, cytokine release syndrome (CRS), and infections occurring without consistent signs of biological activity. These studies revealed several pitfalls associated with BiTEs-based therapy, including a short in vivo half-life requiring continuous intravenous (cIV) infusion to maintain adequate plasma levels and unexpected toxicities related to higher dosages [94]. 

Subsequent studies adopted cIV infusion with a dose-escalating approach, to achieve the target dose while minimizing complications. In the MT103-104 phase I dose-escalation trial, which assessed blinatumomab’s antilymphoma activity in 76 heavily pretreated patients with R/R B-NHLs (mainly R/R FL, R/R MCL, and R/R DLBCL), the maximum tolerated dose (MTD) was identified as 60 μg/m^2^/day. Among thirty-five patients treated at the MTD, the ORR was 69% across all B-NHL subtypes and 55% for DLBCL, with a global mDOR of 404 days [95]. In an FU analysis of thirty-eight patients who participated in the MT103-104 phase I trial, the mOS was 4.6 years, and the mPFS was 6.7 months. Responders had significantly improved outcomes, with an mOS and an mPFS of 7.7 years and 3.2 years, respectively [96]. A phase II using flat-dose blinatumomab in 23 R/R DLCBL patients showed that achieving a response is closely tied toreaching the highest dose. In this study, nearly a third of patients did not reach the highest expected dose (112 μg/day) due to rapid disease progression or AEs, especially neurological ones. Among evaluable patients, blinatumomab demonstrated moderate activity with an ORR of 43%, including CRs in 19% of cases [97]. Similar results were obtained in a phase II–III study evaluating the efficacy of a flat-dose blinatumomab as a second salvage therapy for R/R aNHL following platinum-based salvage chemotherapy. In the phase II portion, 41 patients were enrolled and over half of them discontinued treatment during the first cycle due to disease progression or toxicities. Neurologic grade ≥ 3 AEs occurred in 24% of patients, and CRS grade ≥ 3 in 2%, with ORR and CR rates of 37% and 22%, respectively [98].

To simplify administration, improve convenience, and potentially reduce AEs, SC blinatumomab formulations are being investigated. A phase I study involving SC blinatumomab, following three weeks of cIV infusion in nearly thirty R/R iNHLs patients, showed a favorable safety profile and efficacy comparable to the cIV formulation [99]. 

Blinatumomab has also shown promise as a consolidation therapy. In a multi-center, single-arm phase II study, cIV blinatumomab was evaluated in high-risk DLBCL patients who were not progressing after first-line immunochemotherapy. The treatment was better tolerated compared to previous trials, with 11% G ≥ 3 neurologic AEs and no G ≥ 3 CRS reported. Among 28 enrolled patients, 3 out of 4 with PR and all four with stable disease (SD) after frontline therapy achieved CR [100]. In a recently published pilot study, 14 patients with DLBCL or trFL received cIV blinatumomab, administered according to the B-ALL schedule as consolidation therapy after ASCT. All patients completed treatment, with 86% achieving CR 100 days after ASCT and 50% maintaining the response one year after transplant. The mPFS was 18 months, and the mOS was not reached with an mFU of 37 months. Toxicities were more manageable compared to previous studies [101]. 

#### Blinatumomab in Combined Modalities

In B-NHLs, blinatumomab has also been evaluated in combination with other compounds. In the phase I KEYNOTE-348 study, cIV blinatumomab was combined with the anti-PD1 checkpoint inhibitor pembrolizumab in patients with R/R DLBCL. The trial was terminated after the dose-finding phase due to unacceptable toxicities and limited anti-lymphoma activity [102]. Another phase I trial investigating blinatumomab plus lenalidomide in patients with R/R B-NHLs is underway (NCT02568553). 

## 4. Anti CD30 Antibodies

### 4.1. CD30 × CD16A Targeting

AFM13 is a tetravalent BsAb, or tandem diabody (TandAb), designed to bind CD16A, an isoform primarily expressed on natural killer (NK) cells, that acts as a low affinity receptor for the IgG Fc region, and CD30 on malignant cells [Figure 2]. By simultaneously engaging CD30 and CD16A, AFM13 mediates a potent cytotoxic effect [103,104,105]. However, the antitumor activity of NK cells diminishes in heavily pretreated patients due to cellular exhaustion or nonspecific toxicity induced by prior therapies [105].

In the first-in-human phase I study involving 28 patients affected by R/R cHL, AFM13 was used as a single agent in a dose-escalation scheme ranging from 0.01 mg/Kg to 7 mg/Kg weekly for one month. Modest antitumor activity was observed, with an ORR of 11.5% and no CR. The most common AEs were fever, chills, headache, and nausea, with 9.2% of G ≥ 3 AEs [106]. A subsequent phase II trial conducted by Sasse et al. [107] showed similar results in 25 patients with R/R cHL, reporting an ORR of 16.6% (1 CR, 3 PR, 6 SD) and two serious AEs. Interestingly, in the phase I trial by Rothe et al. [106], the presence of anti-AFM13 antibodies was determined in approximately half of the treated patients.

Despite the limited single-agent efficacy, promising results have been observed when AFM13 is combined with other agents. A phase Ib study evaluated the safety and efficacy of AFM13 in combination with pembrolizumab in heavily pretreated R/R cHL. AFM13 was administered to 30 patients in a dose-escalating scheme alongside a fixed 200 mg flat dose of pembrolizumab. The study showed an ORR of 83%, including 37% CR. Notably, 85% of patients refractory to the anti-CD30 antibody–drug-conjugated brentuximab vedotin responded, with 46% achieving CR. The mDOR was 9.9 months, and the combination demonstrated an acceptable safety profile. The most common AEs were infusion-related reactions, cutaneous rash, nausea, fever, and diarrhea [108].

In a small cohort of 15 patients with R/R cutaneous CD30-positive T-cell non-Hodgkin lymphoma (T-NHLs), AFM13 was administered using four different schemes, resulting in an ORR of 40%. The treatment demonstrated activity even in brentuximab-vedotin-resistant subjects [109]. In another phase II study evaluating AFM13 in patients with CD30-positive peripheral T-cell lymphomas (PTCLs) or transformed mycosis fungoides (NCT04101331), weekly intravenous doses of 200 mg yielded promising outcomes. The mDOR, mPFS, and mOS were 2.3, 3.5, and 13.8 months, respectively. The ORR was different among subgroups with better outcomes for patients with >0.9 × 10^9^ lymphocytes/L and angioimmunoblastic T-cell lymphoma subtype [110].

Autologous NK cells may be compromised in heavily pretreated patients because of nonspecific toxicity induced by chemotherapy or cellular exhaustion determined by compounds that directly stimulate NK cells. To address this limitation, Nieto et al. [111] conducted a phase I–II trial to evaluate AFM13 in combination with preactivated and expanded cord blood-derived NK cells in R/R CD30-positive lymphomas. Forty-two heavily pretreated patients were enrolled and safety analyses demonstrated a low incidence of AEs with none of the subjects experiencing CRS, ICANS, or graft versus host disease (GVHD). At an mFU of 14 months, the event-free survival and OS rates were 31% and 76%, respectively. Nine patients underwent transplant consolidation (five allogeneic stem-cell transplant and four ASCT), with eight remaining event-free for 10–34 months. Building on this rationale, Moskowitz et al. [112] are working on the combination of AFM13 with allogeneic NK cells in R/R cHL and CD30-positive PTCLs in an ongoing and recruiting phase II trial.

### 4.2. CD30 × CD3 Targeting

DuoBody-CD3 × CD30 (GEN3017) is a Fc-silenced IgG1 BsAb under development for the treatment of CD30-positive hematological malignancies, including cHL and anaplastic large-cell lymphoma (ALCL). Preclinical data have shown promising results in in vitro, ex vivo, and in vivo murine studies, with the induction of CD4+ and CD8+ T-cell activation, proliferation, cytokine production, and potent T-cell-mediated cytotoxicity [Figure 2] [113,114]. A first-in-human trial testing GEN3017 in CD30-positive lymphomas is currently ongoing (NCT06018129).

## 5. Other Novel Compounds and Targets

A-319, a CD19 × CD3 BsAb, is under investigation in the Chinese population for the treatment of B-ALL and B-NHLs. Results from the phase I trial in patients with R/R B-NHLs showed a favourable toxicity profile, with no G ≥ 3 CRS and limited neurotoxicity [115].

Englumafusp alfa has shown interesting preliminary results in combination with glofitamab. This antibody-like fusion protein targets CD19 on B cells and 4-1BB on T cells. By targeting 4-1BB, it co-stimulates T cells, enhancing their effector functions and preventing anergy. A phase I study investigating this combination in R/R B-NHLs demonstrated a good toxicity profile, with no additive or synergistic safety concerns compared to glofitamab monotherapy. Promising efficacy was also observed with an ORR and CR rate of 67% and 57%, respectively, among 83 patients with R/R diffuse large B-cell lymphoma (DLBCL), including those previously treated with CAR-T therapies. High rates of minimal residual disease (MRD) negativity were also achieved [116,117].

Amulirafusp alfa (IMM0306) is a CD20 × CD47 fusion protein consisting of a CD20 mAb linked to the CD47-binding domain of SIRPα on both heavy chains [118]. It was tested on R/R B-NHL patients in a phase I study. The safety profile seems manageable while the efficacy results are quite preliminary [119]. CD47 interacts with SIRPalpha, initiating inhibitory signaling, and has recently been labeled as a macrophage checkpoint. CD47 blockage enhances macrophage activity against tumor cells [120].

AZD0486 (formerly TNB-486) is a novel CD19 × CD3 BsAb currently under development. It is a IgG4 T-cell engager with a unique anti-CD3 moiety designed to reduce CRS and ICANS [121,122]. 

Among novel CD20 × CD3 BsAbs, CM355 and JS203 are being investigated in phase I–II trials for R/R B-NHL (NCT05210868 and NCT05618327). Similarly, JNJ-75348780, a novel CD22 × CD3 BsAb, is undergoing a phase II trial for various R/R B-cell malignancies (NCT04540796).

Several innovative mAbs are in development that target three (tri-specific) or four (tetra-specific) different targets. These compounds feature heterogeneous structures comprising different domains and epitopes able to simultaneously bind different targets eliciting an antitumor immune response. For example, JNJ-80948543 is a tri-specific antibody containing anti-CD3 and anti-CD20 single-chain variable fragments, an anti-CD79b fragment antigen-binding domain, and an effector-silent Fc. It is currently under investigation for R/R B-NHLs treatment, both as a monotherapy and combined with other T-cell engagers (NCT05424822 and NCT06139406). CMG1A46, a tri-specific antibody targeting CD3 × CD20 × CD19, is under evaluation for R/R B-NHL and R/R B-ALL (NCT05348889). The tri-specific antibody PIT565 targeting CD3 × CD2 × CD19 is currently being tested as a monotherapy, as an IV or SC formulation in R/R B-NHL and R/R B-ALL (NCT05397496). IPH6501 is an antibody-based NK-cell engager therapeutic (ANKET) consisting in a tetra-specific molecule engaging NKp46 and CD16a on NK cells, CD122 on the interleukin-2 receptor (IL-2R), and CD20. This multiple binding induces NK-cell activation and proliferation and triggers NK cytotoxicity against tumor cells [123]. It is currently being tested in a phase I–II study in CD20-positive B-NHLs (NCT06088654).

GNC-038 is a first-in-class octavalent, CD19 × CD3 × 4-1BB × PD-L1 tetra-specific antibody. It is under investigation in a phase I trial for CD19-positive R/R B-NHLs and R/R B-ALL patients (NCT04606433).

## 6. Conclusions

In recent years, remarkable progress in the immunotherapy of hematologic malignancies has significantly improved outcomes for patients with R/R lymphoid diseases. BsAbs are now a viable and effective therapeutic option, offering well-tolerated and efficacious treatments. These agents are expected to play an important role in managing both frontline and R/R lymphoma patients in the near future, particularly because of their ‘off the shelf’ nature. While new data continue to emerge, especially regarding agents targeting CD19, CD30, and NK cells, challenges remain in determining the optimal sequencing of these therapies in both frontline and R/R settings. As multiple novel therapies are introduced, clinicians will need to address these complexities to maximize patient outcomes.

## Figures and Tables

**Figure 1 cancers-17-00094-f001:**
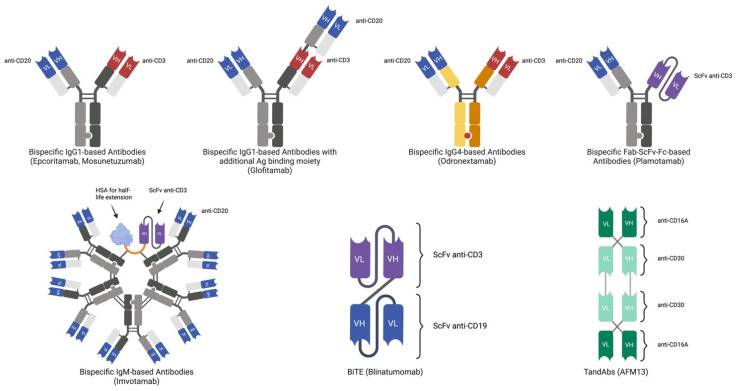
Simplified molecular structure of anti-CD20, anti-CD19, and anti-CD30 bispecific products. Abbreviations: BiTE (bispecific T-cell engager), ScFV (single-chain variable fragment), and TandAbs (tandem diabodies).

**Figure 2 cancers-17-00094-f002:**
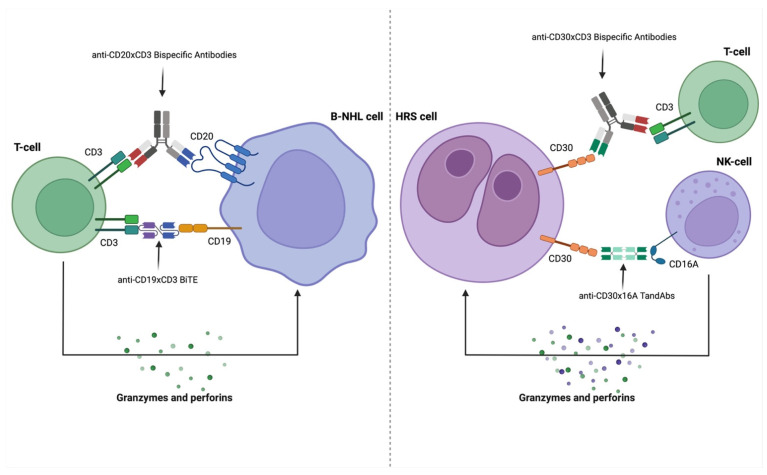
Mechanisms of action and targets of anti-CD20, anti-CD19, and anti-CD30 bispecific products. Abbreviations: BiTE (Bispecific T-cell Engager), HRS cell (Hodgkin and Reed Sternberg cell), NHL cell (Non-Hodgkin Lymphoma cell), and TandAbs (Tandem diabodies).

**Table 2 cancers-17-00094-t002:** Glofitamab, mosunetuzumab, epcoritamab, and odronextamab ongoing clinical trials.

Clinical Trial Number	Condition	Phase	Used Treatment
NCT06050694	1L DLBCL	II	Glo + Pola-R-CHP
NCT06186648	RT	II	R-CHOP + Glo or Obinu-CHOP + Glo
NCT05783596	1L FL or MZL	II	Glo + Obinu
NCT04703686	R/R B-NHL	II	Glo
NCT06192888	R/R MCL	I	Glo + Len
NCT05335018	R/R DLBCL	II	Glo + Poseltinib + Len
NCT04889716	R/R DLBCL or trFL	II	Glo or Mosu after CAR-T
NCT06071871	R/R LBCL	II	Glo + PV + Obinubefore and after CAR-T
NCT06213311	R/R LBCL	II	axicabtagene ciloleucel + Glo
NCT05833763	R/R MCL	II	Glo + Pirtobrutinib
NCT05896163	R/R DLBCL	I-II	Glo + Maplirpacept
NCT05219513	R/R B-NHL	I	Glo + RO7443904
NCT04077723	R/R B-NHL	I-II	RO7227166 + Glo or RO7227166 + Obinu
NCT05798156	1L Aggressive BCL	II	Glo + PV + R
NCT06043674	RT	II	Glo + PV or Glo + Atezolizumab
NCT06047080	1L LBCL	III	Glo + PV + CHP vs. PV-R-CHP
NCT05800366	1L DLBCL	II	Glo + PV-R-CHP
NCT05364424	R/R DLBCL	I	Glo + R-ICE
NCT04914741	1L DLBCL	I-II	Glo + PV-R-CHP vs. Glo + R-CHOP
NCT05169515	R/R B-NHL	I	Glo or Mosu + CC-220 and CC99282
NCT04970901	R/R B-NHL	I	Lonca + Glo or PV or Mosu
NCT06252675	R/R MCL	II	Glo + Pirtobrutinib
NCT06464185	R/R B-NHL	I-II	Glo + CAR-T
NCT05861050	1L MCL	I-II	Glo + Obinu+ Ven + Len
NCT06054776	R/R MCL	II	Glo + Acalabrutinib + Obinu
NCT06357676	1L MCL	I-II	Glo + Ibrutinib + Obinu
NCT04161248	R/R Aggressive BCL	I	Glo + R-GDP vs. Tafasitamab + R-GDP vs. Ven + R-GDP
NCT03467373	1L DLBCL	I	Glo + PV-R-CHP vs. Glo + R-CHOP or Obinu-CHOP
NCT05849857	R/R FL	II	Mosu
NCT05412290	R/R aggressive BCL	I	Mosu after ASCT
NCT05389293	1L FL	II	Mosu
NCT04246086	1L or R/R FL	I-II	Mosu sc or iv + Len
NCT05994235	1L FL	II	Mosu + Tazemetostat
NCT05410418	1L FL	II	Mosu + PV
NCT04792502	1L FL or MZL	II	Mosu + Len
NCT05091424	R/R CLL	I	Mosu + Ven
NCT05464329	R/R aggressive BCL	I	Mosu + Platinum-based salvage CHT
NCT05207670	1l or R/R B-NHL	II	Mosu
NCT03677154	1L DLBCL	I-II	Mosu or Mosu + PV
NCT06492837	R/R FL	II	Mosu + Zanubrutinib
NCT05169658	1L indolent B-NHL	II	Mosu or Mosu + Obinu+ PV
NCT04712097	R/R FL	III	Mosu + Len or R + Len
NCT05615636	R/R FL or DLBCL	II	Mosu + PV + Len + Tafasitamab
NCT05260957	R/R aggressive BCL	II	Mosu + CAR-T + PV
NCT06442475	1L indolent BCL	II	Mosu
NCT05171647	R/R aggressive BCL	III	Mosu +PV or R-GemOx
NCT05672251	R/R DLBCL	II	Mosu + Lonca
NCT06337318	1L FL	III	Mosu vs. R
NCT06015880	R/R DLBCL	I	Mosu + PV + Len
NCT06453044	R/R FL	II	Mosu + PV
NCT06284122	1L FL	III	Mosu + Len vs. antiCD20 + CHT
NCT06006117	R/R MZL	III	Mosu + Len vs. Investigator’s choice
NCT06249191	1L c-myc+ DLBCL or HGBCL	I-II	Mosu + DA-EPOCH
NCT05886036	NLPHL	II	Mosu or R
NCT05633615	R/R DLBCL	II	Mosu + PV or Mosu after CAR-T
NCT06287398	R/R LBCL	II	Epco + CHT
NCT06458439	R/R LBCL	II	Epco before and after CAR-T
NCT05451810	R/R DLBCL or FL	II	Epco
NCT06447376	R/R DLBCL or FL	I	Siltuximab + Epco
NCT06414148	R/R LBCL	II	Epco or Epco + R + Len
NCT05783609	1L FL	II	Epco + R
NCT05660967	1L DLBCL	II	Epco or Epco + Len
NCT05791409	R/R CLL or SLL	I-II	Epco + Ven
NCT04623541	R/R CLL or RT	I-II	Epco
NCT05409066	R/R FL	III	Epco + R + Len
NCT04628494	R/R DLBCL	III	Epco vs. Investigator’s choice
NCT04542824	R/R B-NHL	I-II	Epco
NCT06045247	1L DLBCL	II	Epco + R-miniCVP
NCT05578976	1L DLBCL	III	Epco + R-CHOP vs. R-CHOP
NCT05201248	R/R BCL	I-II	Epco or Epco + SOC
NCT05852717	R/R LBCL	II	Epco + GDP
NCT06191744	1L FL	III	Epco + R2
NCT05848765	R/R FL	II	Epco or CHT
NCT06112847	1L FL	II	Epco + Len
NCT05283720	B-NHL	II	Epco + CHT or Immunotherapy
NCT06238648	R/R aggressive BCL	II	Epco after CAR-T
NCT05685173	R/R aggressive BCL	I	Odro + REGN5837
NCT02651662	R/R aggressive BCL	I	Odro + Cemiplimab
NCT06149286	R/R FL or MZL	III	Odro + Len vs. R + Len
NCT06091865	1L DLBCL	III	Odro + CHOP vs. R-CHOP
NCT06097364	R/R FL	II	Odro + CHOP vs. R-CHOP
NCT06091254	1L FL	III	Odro vs. R-CHOP or R-CVP or BR
NCT06230224	R/R aggressive BCL	III	Odro vs. SoC

Abbreviations: 1L (First-Line), ASCT (Autologous Stem-Cell Transplant), BCL (B-Cell Lymphoma), B-NHL (B-Cell Non-Hodgkin Lymphoma), BR (Bendamustine and Rituximab), CAR-T (Chimeric Antigen Receptor T-Cells), CHOP (Cyclophosphamide, Doxorubicin, Vincristine, and Prednisone), CHP (Cyclophosphamide, Doxorubicin, and Prednisone), CHT (Chemotherapy), CLL (Chronic Lymphocitic Leukemia), CVP (Cyclophosphamide Vincristine and Prednisone), DA-EPOCH (Dose-Adjusted Etoposide, Prednisone, Vincristine, Cyclophosphamide, and Doxorubicin), DLBCL (Diffuse Large B-Cell Lymphoma), Epco (Epcoritamab), FL (Follicular Lymphoma), Obinu (Obinutuzumab), GDP (Gemcitabine, Dexamethasone, and Cisplatin), GemOx (Gemcitabine and Oxaliplatin), Glo (Glofitamab), HGBCL (High-Grade B-Cell Lymphoma), ICE (Ifosfamide, Carboplatin, and Etoposide), iv (Intravenous), LBCL (Large B-Cell Lymphoma), Len (Lenalidomide), Lonca (Loncastuximab Tesirine), MCL (Mantle Cell Lymphoma), Mosu (Mosunetuzumab), MZL (Marginal Zone Lymphoma), NLPHL (Nodular Lymphocyte-Predominant Hodgkin Lymphoma), Odro (Odronextamab), pola-R-CHP (Polatuzumab Vedotin, Rituximab, Cyclophosphamide, Doxorubicin, and Prednisone), PV (Polatuzumab Vedotin), R (Rituximab), R/R (Relapsed or Refractory), R2 (Rituximab and Lenalidomide), RT (Richter’s Transformation), sc (Subcutaneous), SLL (Small Lymphocitic Lymphoma), SoC (Standard of Care), trFL (Transformed Follicular Lymphoma), and Ven (Venetoclax).

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
