# Peer review of "Bispecific Antibodies for Lymphoid Malignancy Treatment"

_cancers, 2024, doi:10.3390/cancers17010094_

Round 1
Reviewer 1 Report
Comments and Suggestions for Authors
The review is well written, current and comprehensive. As there was considerable abbreviations used throughout the text, the reader would be much aided by adding an abbreviations referral table.
The authors may wish to check possible grammar errors/ word choice on lines 94, 166 and 541.
As a suggestion please define "silenced crystallizable fragment" on line 66.
Author Response
Comment 1 The authors may wish to check possible grammar errors/ word choice on lines 94, 166 and 541
Response: grammars errors corrected
Comment 2 As a suggestion please define "silenced crystallizable fragment" on line 66.
Response: It has strong affinity for CD20, prolonged half-life, and silenced crystallizable fragment (Fc) which consists in an altered Fc region unable to mediate effector functions
Reviewer 2 Report
Comments and Suggestions for Authors
Line 88:
mean duration of response (mDOR) of 18.4 months – median?
Table 1
I would suggest using “obinutuzumab” instead of “Ga101” (also in other paragraphs/ tables)
The word epcoritamab is cut to epcorit
What about the indication for epcoritamab in DLBCL?
Lines 115-117
“in the latter were observed reduced CRS rates and severity with G 4 CRS cases only…” – sentence editing required
Table 2
“CVP (Ciclofosfamide, ….” – please correct
Lines 194-6
“In another report has been highlighted that major part of infections were not related with neutropenia and focused during the first 4 cycles.” - sentence editing required
Line 204
Educed transcription – please correct
Line 223
Hematopoietic stem-cell transplant
Line 244 – English editing required
Line 260: TP53 mutations – please use italics for the genes
Epcoritamab – please update the information on approvals by FDA and EMA
Line 337, lines 345-6, 421-2, 462-4 – English editing
Line 481 – please provide details on transplant consolidation – ASCT?
Author Response
Comment 1 Line 88: mean duration of response (mDOR) of 18.4 months – median?
Response: Glofitamab single agent revealed consistent efficacy in heavily pretreated DLBCL, obtaining complete response (CR) rates of 36.8-39% and overall response rates (ORR) of 52-53.8%, median duration of response (mDOR) of 18.4 months and median duration of CR (mDOCR) of 26.9 months
Comment 2 I would suggest using “obinutuzumab” instead of “Ga101” (also in other paragraphs/ tables) The word epcoritamab is cut to epcorit What about the indication for epcoritamab in DLBCL
Reponse: corrected in the table with indication for Epcoritamab in DLBCL
Comment 3: Lines 115-117“in the latter were observed reduced CRS rates and severity with G 4 CRS cases only…” – sentence editing required
Response: The cohort exposed to a higher dose of Obinutuzumab experienced reduced CRS rates and fewer cases of G 4 CRS
Comment 4: table 2 “CVP (Ciclofosfamide, ….” – please correct
Response: corrected
Comment 5 Lines 194-6 “In another report has been highlighted that major part of infections were not related with neutropenia and focused during the first 4 cycles.” - sentence editing required
Response: In another report has been highlighted that most infections were not related with neutropenia and more frequent during the first 4 cycles
Comment 6: Educed transcription – please correct
Response: corrected
Comment 7 Hematopoietic stem-cell transplant
Reponse: corrected
Comment 8: Line 244 – English editing required
Reponse: The best ORR was 87.5% with 85% of CR and a 2-year PFS of 65%. In this study the mDOR was not reached while the 2-year DOCR was 70.9% considering a mFU of 32 months
Comment 9: Line 260: TP53 mutations – please use italics for the genes
Reponse: corrected
Comment 10: Epcoritamab – please update the information on approvals by FDA and EMA
Reponse: Epcoritamab is currently approved by FDA and EMA2 for treatment of R/R FL and R/R DLBCL
Comment 11: Line 337, lines 345-6, 421-2, 462-4 – English editing
Response:The ORR and CR rate for the whole cohort were 51% and 37%, respectively, 91% and 72% in FL, 48% and 30% in DLBCL after CAR-T and 53% (all CR) in DLBCL not previously exposed to CAR-T [81,82]. The Phase II ELM-2 study confirmed good responses in R/R DLBCL with a longer mFU of 26.2 months: 52% ORR (66/127), 31% CR (39/127) and mDOCR of 17.9 months [83]. An interim analysis on the R/R FL cohort of the same study confirmed great efficacy in this subset: 80% of ORR (102/128) with 72% of CR (92/128), a mPFS of 26.6 months and a mDOCR of 21.7 months [84]. In both groups the 2-year mDOCR was 48%. The safety profile of odronextamab is comparable to other BsAbs:themost common AEs were infections, hematological toxicities, ICANS and CRS. The last two were apparently more severe compared to other compounds. CRS occurred in 61% of patients with G ≥3 in 7%
Comment 12 Line 481 – please provide details on transplant consolidation – ASCT?
Response:At a mFU of 14 months the event-free survival and OS rates were 31% and 76%, respectively; 9 patients underwent a transplant consolidation (5 allogeneic stem-cell transplant and 4 ASCT) and 8 of them remained event-free at 10-34 months
Reviewer 3 Report
Comments and Suggestions for Authors
The review by Matteo Bisio et al. offers a highly detailed and timely overview of recent trials involving bispecific antibodies (BsAbs) in patients with relapsed/refractory (R/R) B-cell non-Hodgkin lymphoma (B-NHL) and Hodgkin’s lymphoma. It covers single-agent outcomes, emerging combination therapies, new BsAb constructs, and associated toxicities, making it a comprehensive resource on this rapidly evolving area.
The review is well-written and successfully synthesizes an extensive amount of data, presented effectively through the text and numerous tables. However, one of the main limitations of the paper is the lack of guidance on which therapeutic options appear most promising, particularly regarding combination therapies. Including a table that compares the efficacy and toxicity of key BsAbs and their combinations, along with a brief discussion highlighting the authors' preferred options, would enhance the paper’s value to readers.
In most cases, the authors provide detailed descriptions of antibody formats with the aid of Figure 1. However, additional information on the Fc regions of different antibodies—including isotype, modifications (such as silencing), and their roles in either activating or blocking biological effector functions—would be beneficial for understanding the mechanisms behind these therapies.
Specific points:
Line 59. What is functional characteristics has the IgG1 used in this antibody (silences, ADCC or complement activation)?
Line 155. Explain with more detail the format and functional characteristics of Mapliparcept “anti-CD47 antibody-like fusion protein”.
Line 161. It is mentioned that Moseunetuzumab is a humanized Ig1-based antibody that lack complement activation, but what about the other effector functions?
Line 264. Could the author present more information about the epitope differences between this antibody and the other CD20s?
Line 361. Could the authors discuss the issues related to the half-life and the affinity/avidity of this antibody format?
Line: 444. Can the authors a reason for the statement in lines 44-445 about the antitumoral activity of the NK decrease in heavily treated patients?
Line 509: Describe the format of this CD20xCD47 fusion protein.
Lines 523 and 533: Please, describe the formats of these to antibodies.
Author Response
Comment 1 One of the main limitations of the paper is the lack of guidance on which therapeutic options appear most promising, particularly regarding combination therapies. Including a table that compares the efficacy and toxicity of key BsAbs and their combinations, along with a brief discussion highlighting the authors' preferred options, would enhance the paper’s value to readers.
Response: thank you for the revision and the interesting comments. The main topic of our revision was a complete revision of all the therapeutic scenario with bispecific antibodies in lymphoma, both in LBCL and indolent. Actually a lot of clinical trials, alone or in combination, are testing bispecific antibodies in different settings, frontline or in relapse, and we are not able to have a defined preferred option. For this reason our review is oriented in a synthesis of all possible options
Comment 2 In most cases, the authors provide detailed descriptions of antibody formats with the aid of Figure 1. However, additional information on the Fc regions of different antibodies—including isotype, modifications (such as silencing), and their roles in either activating or blocking biological effector functions—would be beneficial for understanding the mechanisms behind these therapies.
Reponse: thank you for the comment. We have revised and added informations on the Fc regions in the text
Comment 3 Line 59. What is functional characteristics has the IgG1 used in this antibody (silences, ADCC or complement activation)?
Reponse: It has strong affinity for CD20, prolonged half-life, and silenced crystallizable fragment (Fc) which consists in an altered Fc region unable to mediate effector functions [14]. Its elevated binding affinity theoretically grants good efficacy in combination strategies or in sequential therapies with other monoclonal antibodies (mAbs).
Comment 4: Line 155. Explain with more detail the format and functional characteristics of Mapliparcept “anti-CD47 antibody-like fusion protein”.
Response: Mapliparcept is an antibody-like fusion composed by a binding domain for CD47, derived from SIRPalfa, linked to an Fc domain 3that enhances phagocytosis and antitumor activity via inhibition of CD47-mediated signaling
Comment 6: Line 161. It is mentioned that Moseunetuzumab is a humanized Ig1-based antibody that lack complement activation, but what about the other effector functions?
Response: Mosunetuzumab is a humanized IgG1-based CD20xCD3 BsAb [Figure 1] with an altered Fc that does not bind complement or Fc gamma receptor. It has a single rituximab-like binding site to CD20 and a single binding site to CD3. Similarly to other CD20xCD3 BsAbs the first cycle of mosunetuzumab impose a dose escalation schedule conceived to reduce CRS incidence
Comment 7 Line 264. Could the author present more information about the epitope differences between this antibody and the other CD20s?
Response:Epcoritamab is a CD20xCD3 IgG1 full-length BsAb for SC administration [Figure 1] [62–64]. Utilizing an ofatumumab-like domain, it binds a specific site of CD20 which differs from other approved CD20-targeting BsAbs since it is able to bind both large and small extracellular loop of the CD20 protein
Comment 8: Line: 444. Can the authors a reason for the statement in lines 44-445 about the antitumoral activity of the NK decrease in heavily treated patients?
Response: Autologous NK cells may be dysfunctional in patients exposed to multiple agents because of nonspecific toxicity induced by chemotherapy or cellular exhaustion determined by compounds that directly stimulate NK cells. For this reason antineoplastic function of AFM13 could be undermined in the setting of R/R patients
Comment 9: Line 509: Describe the format of this CD20xCD47 fusion protein.
Response: Amulirafusp alfa (IMM0306) is a CD20xCD47 fusion protein composed of a CD20 mAb with the CD47 binding domain of SIRPα on both heavy chains
Comment 10: Lines 523 and 533: Please, describe the formats of these to antibodies.
Response: These compounds show heterogeneous structures comprising different domains and epitopes able to bind simultaneously different targets eliciting an antitumoral immune response. For example, JNJ-80948543 is a tri-specific antibody comprising an anti-CD3 and anti-CD20 single-chain variable fragments, an anti-CD79b fragment antigen-binding domain and an effector-silent Fc.
Reviewer 4 Report
Comments and Suggestions for Authors
M Bisio et al. have written a review article on the topic of bispecific antibodies for treatment of lymphoid malignancies. The paper is well written and the reader can easily understand its content. My main concern is that the article is merely descriptive of the bispecific antibodies available, explaining the different indications and clinical trials performed using them. The authors should have been discussed critically their use and compare this modality of treatment regarding effectivity and toxicity with others, such as CAR-T cells and stem cell transplantation among others. Another concern is that the authors do not show any personal experience published in this field. Another minor comments:
- Abstract: Hodgking disease is an old-fashion term. Hodgkin lymphoma is more appropriate nowadays.
- Introduction. A brief history of the development and first steps in the use of bispecific antibodies in the treatment of lymphomas would enrich this section and the article.
References 1 and 2 are not appropriate for the sentence in which are placed as they are references about first line treatment rather than to R/R NHL. Additionally, these references are too old.
- Figure 1. Letters are too small.
- Many abbreviations have not been defined. Some examples: CTCAE in line 80; mPFS in line 169, mOS in line 170m mDOR in line 177, mDOCR in line 177, mFU in line 333. Explain iNHL and aNHL.
- Table 1. The superscripts on the left column of the table should be explained at the bottom. References at the bottom should be indicated in the table if this is their purpose. Ga101 has not been defined.
- Line 204: please clarify or correct “educed transcription”. Do you mean “reduced transcription”?
- The terms “T-cell engager” and “bispecific antibody” seem to be used indistinctly in the article with the same meaning. The authors should use only one of them if they are used indistinctly or explain which is the difference between one and the other term.
Author Response
Comment 1: The authors should have been discussed critically their use and compare this modality of treatment regarding effectivity and toxicity with others, such as CAR-T cells and stem cell transplantation among others.The authors do not show any personal experience published in this field.
Response: thank you for the complete revision and the suggestions. Our paper was dedicated to a complete revision of the possibile treatments with bispecific antibodies in lymphoid malignancies both in aggressive and indolent. Actually we have limited personal experience with these treatments.
Comment 2 Abstract: Hodgking disease is an old-fashion term. Hodgkin lymphoma is more appropriate nowadays.
Response: updated
Comment 3 Introduction. A brief history of the development and first steps in the use of bispecific antibodies in the treatment of lymphomas would enrich this section and the article.
Response: Introduction was revised
Comment 4: References 1 and 2 are not appropriate for the sentence in which are placed as they are references about first line treatment rather than to R/R NHL. Additionally, these references are too old.
Response: references were revised
Comment 5: - Figure 1. Letters are too small.
Response: updated
Comment 6: Many abbreviations have not been defined. Some examples: CTCAE in line 80; mPFS in line 169, mOS in line 170m mDOR in line 177, mDOCR in line 177, mFU in line 333. Explain iNHL and aNHL.
Response: we updated the draft with a complete table of all abbreviations
Comment 7: Table 1. The superscripts on the left column of the table should be explained at the bottom. References at the bottom should be indicated in the table if this is their purpose. Ga101 has not been defined.
Response: updated
Comment 8: - Line 204: please clarify or correct “educed transcription”. Do you mean “reduced transcription”?
Response: updated
comment 9: The terms “T-cell engager” and “bispecific antibody” seem to be used indistinctly in the article with the same meaning. The authors should use only one of them if they are used indistinctly or explain which is the difference between one and the other term.
Reponse: updated with Bispecific antibodies (BsAbs)
Round 2
Reviewer 4 Report
Comments and Suggestions for Authors
M Bisio et al reviewed the manuscript. The article in general is merely descriptive and it lacks of a critical view of the use of the different bispecific antibodies and their impact on the treatment of lymphoid malignancies both, in the present and in the near future. A discussion comparing with other available treatments, such as CAR-T cell therapy and other types of novel antibodies, is also missing.
Specific comments:
- The introduction does not give a complete background of the use of bispecific antibodies in lymphoid malignancies. The authors should include a brief history of the use of these new antibodies, explaining when they first were included in clinical practice and what are the currently approved indications in the treatment of lymphoid malignancies.
- Letters in figure 1 appear a little blurred.
- A table with abbreviations has been included rather than writing their definitions when they appear first in the text, which is weird in an article.
Author Response
Comment 1: The introduction does not give a complete background of the use of bispecific antibodies in lymphoid malignancies. The authors should include a brief history of the use of these new antibodies, explaining when they first were included in clinical practice and what are the currently approved indications in the treatment of lymphoid malignancies.
Response: Introduction was updated with a brief history concerning the use of CAR-T and bispecific antibodies in lymphoid malignancies. The current approved indications were uploaded in the text for each antibody and in the references in table 1
Comment 2: Letters in figure 1 appear a little blurred.
Response: Figure was uploaded in a different format in order to have clear letters
Comment 3: A table with abbreviations has been included rather than writing their definitions when they appear first in the text, which is weird in an article.
Response: abbreviations were all checked and defined in the text when they appeared first and a complete table was also added